# MicroRNA-Based Fingerprinting of Cervical Lesions and Cancer

**DOI:** 10.3390/jcm9113668

**Published:** 2020-11-15

**Authors:** Justyna Pisarska, Katarzyna Baldy-Chudzik

**Affiliations:** Department of Microbiology and Molecular Biology, Collegium Medicum, University of Zielona Gora, 65-561 Zielona Gora, Poland; k.baldy-chudzik@cm.uz.zgora.pl

**Keywords:** cervical cancer, cervical neoplasia, HPV, microRNA biomarkers, miRNA, molecular diagnostics, prognostic factors

## Abstract

The regulatory functions of microRNA (miRNA) are involved in all processes contributing to carcinogenesis and response to viral infections. Cervical cancer in most cases is caused by the persistence of high-risk human papillomavirus (HR-HPV) infection. While oncogenic human papillomaviruses induce aberrant expression of many cellular miRNAs, this dysregulation could be harnessed as a marker in early diagnosis of HR-HPV infection, cervical squamous intraepithelial lesions, and cancer. In recent years, growing data indicate that miRNAs show specific patterns at various stages of cervical pathology. The aim of this review is to systematize current reports on miRNA capacity that can be utilized in personalized diagnostics of cervical precancerous and cancerous lesions. The analysis of the resources available in online databases (National Center for Biotechnology Information—NCBI, PubMed, ScienceDirect, Scopus) was performed. To date, no standardized diagnostic algorithm using the miRNA pattern in cervical pathology has been defined. However, the high sensitivity and specificity of the reported assays gives hope for the development of non-invasive diagnostic tests that take into account the heterogeneity of tumor-related changes. Due to this variability resulting in difficult to predict clinical outcomes, precise molecular tools are needed to improve the diagnostic and therapeutic process.

## 1. Introduction

Cancer is a group of diseases that in the age of molecular biology development is regarded as the genome disease at the cellular level. It arises from one cell containing an initial set of mutations, and the accumulation of mutations over time with genome instability is common. This leads to intra-tumor genetic heterogeneity characteristic for all types of cancer [1]. The hallmarks of carcinogenesis include cell functional changes: the ability to produce growth factors and lack of sensitivity to their suppressors, inhibition of apoptosis, unlimited replication potential, angiogenesis, invasion, metastasis, changed stress response, metabolic rewiring, and immune modulation. The significant factor contributing to the development of the disease, especially for the cancer caused by infectious agents, is long-lasting inflammation [2,3,4].

Human papillomavirus (HPV) is the second infectious agent in terms of the frequency of inducing oncogenesis in the world [5], and remains a causative agent of cervical, vulvar, anal canal, penile, and head and neck malignancy [6,7]. Cervical cancer (CC) is the fourth most common cancer among women worldwide [8], and high-risk human papillomavirus (HR-HPV) DNA is detectable in over 90% of affected tissues [9]. CC most frequently originates from squamous epithelium (squamous cell carcinoma—SCC) or glandular tissue (adenocarcinoma—AdC). Regardless of the histological type, CC is often asymptomatic and predominantly diagnosed in advanced stage, despite of the occurrence of preceding stages (cervical intraepithelial neoplasia—CIN 1-3) and their long-term development. Late diagnosis limits the effectiveness of available therapeutic methods, including surgical treatment, and chemo- and radiotherapy. Metastatic CC remains incurable and is characterized by 16% 5-year survival rates [10].

CC is preceded by several consecutive cervical intraepithelial lesions that are contributed to HPV-induced persistent infection [11]. The key event associated with the progression to high-grade and tumorigenic changes is the integration of viral DNA into host cell genetic material. This leads to over-expression of viral oncogenes that affect genetic instability and uncontrolled cell cycle progression [12]. Host genetic factors, including alterations in levels of oncogenes and tumor suppressors as well as chromosome aberrations and mentioned lifestyle factors have additional influence on disease development [13,14].

Since the 1950s, the gold standard in the screening of cervical intraepithelial lesions was cytological examination (Pap smear). Due to low sensitivity of these assays, an HR-HPV based preventive system (or co-testing with cytology) has been introduced in developed countries [15]. It was proved to be more effective for early detection of AdC, the frequency of which demonstrates a growing trend in highly developed countries [16]. It was estimated that the proportion of AdCs in relation to SCCs increases from 10% up to 25% of CC cases and becomes more common in younger women [17,18].

In the last decade, research based on explaining the molecular basis of oncogenesis and its utility in diagnostics has been extensively provided. Intensive investigations into the progression of tumorigenic changes are focused on the non-coding regions of the human genome. The central dogma of molecular biology assumes that genetic information transfers from DNA to RNA and then from RNA to proteins in transcription and translation processes. Although 70% of the genome can be transcribed into RNA, the protein-coding sequences represent at most 2% of the human genome [19]. It turned out that nonprotein-coding regions of the genome exhibit crucial regulatory functions involved in the cell and tissue homeostasis. They are transcribed into the non-coding RNAs (ncRNA) from exon, intron, or intergenic sequences and play a key role in the post-transcriptional regulation of gene expression [20]. The most widely studied ncRNAs are microRNAs (miRNA)—RNA molecules up to 25 nt length that strongly regulate the expression of the hundreds target genes at the post-transcriptional level. miRNA affects mRNA by complete or partial hybridizing the seed sequence to the 3′ end of the untranslated region (3′UTR) of the target mRNA. It results in degradation of mRNA or block of translation [21].

It is well documented that miRNAs play an important role in processes that correspond to all the hallmarks of cancer [22]. Well confirmed variability of miRNA expression in a wide range of tumors has revealed significant correlations with the risk of cancer development, advancement of disease, metastasis ability, and therapeutic response to chemo- and radiotherapy [23,24]. Furthermore, in the cases of cervical pathology, specific aberrations in miRNA levels are characteristic for each stage of neoplasia and cancer [25].

Commonly used primary screening assays (exfoliative cytology and HR-HPV) are insufficient to detection of viral persistency and present small predictive value, especially in the risk prognosis of CIN 3 progression to cancer. Furthermore, inconclusive results of screening tests are effective in overusing of invasive diagnostic methods including colposcopy-directed biopsy to perform histological classification of changes [26]. Therefore, it is necessary to develop more efficient solutions, search for new diagnostic approaches to detect precancerous stages, and assess prognostic factors in advanced intraepithelial lesions and cancer.

miRNA pattern is considered as a specific fingerprint of cellular condition and a promising tool with great capabilities in development of personalized medicine. Advances of individualized diagnostics is particularly important in the case of tumors, due to the high heterogeneity even within the same histological type. To date, there has not been an algorithm established that incorporates the alterations of miRNA expression to CC screening. This review is focused on the current state of knowledge about the diversity of potential applications of human miRNA patterns in modern diagnostics at various stages of cervical lesions related to HPV infection and its progression to cancer.

## 2. Pathogenesis of Cervical Cancer and Its Precursor Stages

### 2.1. Histological Subtypes

The cervix contains two types of epithelium: stratified non-keratinizing squamous (ectocervix) and columnar—including glandular cells (endocervix). Both types are joined in squamocolumnar junction (SCJ), called the transformation zone. SCJ is the place where most neoplastic and cancerous lesions originate. It has been described that cells prone to HPV infection are highly consistent with the phenotype observed in high grade precancerous lesions and SCC. They display a characteristic cuboidal columnar phenotype with embryonic origin characterized by specific gene expression [27]. Subsequent studies have shown that these SC junctional cells may be a source of HPV persistency, CINs, and cancer [28]. However, HPV persistency may be involved in all of the cervical sites; infection in glandular cells located close to the metaplasia zone is associated with a higher risk for neoplasia development [29]. The classification of cervical neoplasia is based on histological assessment performed in colposcopy-directed biopsy samples. Abnormal results of cytological assays do not have direct counterparts in histological diagnoses of cervical squamous intraepithelial neoplasia (CIN). Low-grade neoplasia (CIN 1) refers to mild dysplasia that involves up to one-third of the cervical epithelium thickness. About 60% of mild dysplasia is reversible. CIN 2 includes medium-grade dysplastic lesions in one-third to two-thirds of the epithelial layer. The dysplasia affecting more than two-thirds of the epithelium thickness is classified as high-grade dysplasia (CIN 3). CIN 2 and CIN 3 are regarded as direct CC precursors [30].

The majority of cervical cancer cases develop from squamous epithelial tissue —SCC, and from glandular tissue AdC. Low percentage of CC cases represents adenosquamous carcinoma. The epithelial cancers were divided into groups, which included SCC. The most prevalent is large cell cancer (non-keratinizing and keratinizing) and other epithelial cancers, i.e., neuroendocrine cancer [31]. On the basis of histological features, the most prevalent AdC is HPV-dependent mucinous carcinoma. Rare non-mucinous subtypes seem to have less relationship with HPV infection [32,33]. Histological diagnosis is strongly associated with surveillance and susceptibility for treatment. AdC is characterized by worse prognosis relative to SCC. Lymph node involvement or distant metastases occur more frequently and resistance for available treatment methods is common [34].

Strong molecular heterogeneity of CC is related to diversified expression levels of phosphorylated growth factors and cell cycle regulators in different histological subtypes, which promote variable kinases activation. These processes underlie of changes in metabolic signaling pathways and differential activating profiles between SCCs and AdCs [35].

Cervical carcinogenesis is mostly associated with dysregulation of Apolipoprotein B mRNA Editing Catalytic Polypeptide-like (APOBEC)_cytidine deaminases family members [35,36]. It is likely the main source of mutations that have also been described in other tumors [37,38]. According to The Cancer Genome Atlas, mutations occur in CC with a density of about four per megabase on average. The most prevalent genomic alterations in cervical cancers involves PI3K/MAPK (phosphoinositide 3-kinase/mitogen-activated protein kinase) and TGF-β (transforming growth factor beta) signaling pathways. It was estimated that mutations in one or both of them occur in 70% of CC cases. Moreover, mutations observed in genes TGFBR2 (transforming growth factor beta receptor 2), MAPK1 (mitogen-activated protein kinase 1), SHKBP1 (SH3KBP1-binding protein 1), HLA-A (major histocompatibility complex, class I, A), HLA-B (major histocompatibility complex, class I, B), and NFE2L2 (nuclear factor, erythroid 2 like 2) are strongly correlated with SCC. Aberrations in the HLA-A gene in AdC cases was not observed [35]. Differential expressed genes (DEGs) identified in patients may turn out the potential targets for immunotherapy, which is an important alternative to available treatment methods.

### 2.2. HPV as an Infectious Agent

#### 2.2.1. Characteristics

HPV is double-stranded DNA virus classified as part of the Papillomaviridae family, and is infectious to humans and animals. More than 200 types of HPV have been identified to date. On the basis of their DNA sequence, biological function, and pathological effect, they are classified into several genera—alpha, beta, gamma, mu, and nu [39]. Alphapapillomaviruses exhibit epithelial tropism. Of these, a high-risk oncogenic group (HR-HPV) has been defined, including HPV-16, -18, -31, -33, -35, -39, -45, -51, -52, -56, -58, and -59 [40]. The infection initially leads to low-grade intraepithelial lesions that represent slightly changed differentiation patterns. Infected cells in the majority of cases are removed by the immune system within 1 year, likely associated with increased Langerhans cells and vaginal microbiota [41]. Some of low-grade lesions do not spontaneously regress and infection persists for many years, which is closely correlated with the risk of CC development [42]. Persistent HR-HPV infection is asymptomatic and characterized by immune evasion—in particular by interferon pathway that leads to block host’s T cell response [43]. HPV-16 and HPV-18 are the most oncogenic and their presence is identified in over 70% of SCCs and 90% of HPV-positive AdCs [44]. HPV-16 represents several intra-type variants (European, African, and Asian lineages) that are associated with risk of neoplasia developing and histological type of cancer. Accordingly, non-European lineages are more pathogenic and are responsible for more cases of AdC than SCC [45,46]. HPV infection is probably not sufficient to induce neoplastic progression, as evidenced by a low proportion of CCs that have no association with viral infection [47].

#### 2.2.2. Viral Integration into Host Genome and Cell Cycle Affecting

The molecular mechanism of HPV contribution in inducing (pre)cancerous lesions is multifactorial and includes the complex of viral–host interactions. The HPV genome contains regulatory long control region (LCR) with two functional regions consisting of open reading frames (ORFs) for early and late genes [6]. The virus may exist in the host cell nucleus in two physical states—episomal and integrated. Predominantly, both states are observed in HPV-positive cervical tissues simultaneously [48]. The integration process occurs within the early stage of infection and is associated with the progression from low-grade to high-grade intraepithelial lesions. The frequency of total integrations increases with their advancement [48,49,50]. The factors favoring the integration process are correlated with unrepaired damage to host DNA. The DNA damage response (DDR) pathways, which play a key role in genome integrity maintenance, are modified by multidirectional interactions with HPV oncogenes. HPV does not have its own polymerase and employs DDR machinery for replication of the viral genome by involving ataxia telangiectasia-mutated (ATM) and ataxia telangiectasia and Rad3-related (ATR) proteins pathways corresponding to DNA double-stranded break reparation and replication stress response, respectively [51,52].

The detailed mechanism of integration is still the subject of discussion. However, microhomology regions from two to six nucleotides of length between viral/host genome at (or near) the area of integration breakpoints has been noticed in several studies [53,54]. It provides evidence that the theory based on microhomology-mediated DNA repair pathways [55] supported by local genomic instability associated with accumulation of chromosomal changes at the fragile sites is the most probable. Viral DNA may integrate into all chromosomes, as well as intronic and exonic sequences. This process does not occur randomly due to the fact that hotspot genes with an increased frequency of integration sites was identified in independent studies (i.e., tumor-suppressive fragile histidine triad gene—FHIT, or transcriptional factor Kruppel-like factor 5—KLF5) [53,56]. The majority of integration sites are transcriptionally active, however, different patterns of DNA and RNA integration breakpoints have been noticed [56]. Importantly, expression of viral-cellular transcripts increases stability of HPV oncogenes [57,58]. HPV integrations exhibit features of enhancers or activators of flanking genes that can act on their target genes over long distances [59,60]. As described recently, the HPV integration site may contribute to metastasis ability in advanced stage of cancer [61].

Integration events mostly proceed with the viral E1 or E2 ORFs disruption; however, E6 and E7 ORFs always remain unimpaired. This leads to overexpression of E6 and E7 oncogenes or destruction of cellular transcripts [53]. Increased expression of oncogenes may be also caused by genetic and epigenetic LCR modifications in episomal state [62]. Interestingly, recent analysis of DEGs revealed that among HPV positive CCs, integrated and episomal cases must be considered as individual molecular subtypes [63].

E6 and E7 accumulation contributes to cell cycle modifications by affecting key tumor suppressors and chromosomal rearrangements, leading to genome instability [54]. Deregulation of oncogene expression mainly contributes to the disruption of the G1/S cell cycle checkpoint and degradation of fundamental tumor suppressor proteins p53 and pRB (Retinoblastoma protein) by ubiquitin-dependent proteolysis. The decrease in pRB and p53 levels respectively results in bypassing the cell G1 checkpoint and entrance to the S phase, directing cellular processes to allow further viral replication and suppression of apoptosis. Reduced pRB level is associated with releasing E2F transcription factor from suppressive transcriptional RB–E2F complexes. E2F enhances minichromosome maintenance complex component 7 (MCM7) level, which is suppressive for cyclin D expression by associating cyclin-dependent kinase 4—CDK4 [64]. This also leads to reduction of the suppressive effect of inhibitors that affect cyclin-dependent kinase 4 and 6—CDK4/6, such as p16INK4a. It causes cellular accumulation of p16INK4a by activation of a negative feedback loop that is utilized as a high-grade cervical neoplasia marker [65].

In summary, persistency of HPV infection, due to the multidirectional actions, significantly impairs cell functioning, which leads to the accumulation of adverse changes and inhibition of apoptosis.

## 3. miRNA and Cervical Lesions and Cancer

### 3.1. miRNA Biogenesis, Function, and Expression—Modulating Factors

In the last decade, differential miRNA expression profiles have been identified as the possible novel biomarkers in oncology. In the dominant miRNA biogenesis pathway, primary miRNAs (pri-miRNA) are transcribed in the nucleus by polymerase II. Then, pri-miRNA is cleaved into 60–70 nucleotide hairpin precursor (pre-miRNA) by Drosha nucleases. This nuclear processing is followed by the transport of pre-miRNA from the nucleus into the cytoplasm, which takes place through exportin-5 and then by the further processing for mature miRNA by Dicer complexes. Mature miRNA is incorporated into an effector complex—RNA-induced silencing complex (RISC)—which binds to mRNA and affects the translation and stability of mRNA. Expression of the target mRNA is regulated, either by mRNA cleavage or by translational repression, depending on the complementarity of “seed” sequences [66]. Changed miRNA expression is observed in a wide range of cancers versus normal tissues [67,68,69,70,71]. Although altered miRNA profiles in cancer has been established, it is yet to be explained as to whether it is a causative factor or a consequence of malignancy progression.

miRNA regulates crucial processes related to oncogenesis, i.e., stem cell differentiation and maturation, tissue development, cell proliferation, differentiation, growth, survival, invasion, and metastization ability [72]. These molecules are also involved with maintaining metabolic homeostasis and the response to viral infections [73]. Altering expression can lead to it going up or down. miRNAs play a twofold role as tumor suppressors (downregulated in cancer) or oncogenes (oncomiRs, upregulated in cancer). Likewise, miRNAs can act as oncogenes and tumor suppressors simultaneously through interactions in various signal networks [74]. Some of them may influence occurrence of metastasis (metastamiRs) [23,75]. miRNAs exist intracellularly or may be secreted outside the cell in exosomes that play an important role in intercellular communication of cancer cells. As a consequence, repression of target genes in recipient cells and increasing tumor growth and invasiveness is observed. Moreover, exosomal miRNAs are detectable in body fluids, which creates the prospect of developing less invasive diagnostic tests for cancer markers [76,77].

In CC, decrease in tumor-suppressive miRNAs is associated with several factors. Commonly observed in cancer chromosomal aberrations trigger miRNA alterations by a mutational way linked to miRNA loci. Genetic aberrations affect the Drosha expression, whereby the global miRNA profile in CC may be disturbed. A large impact for carcinogenesis was established for hypermethylation of miRNA loci [78]. A particularly important factor of the carcinogenesis process is single nucleotide polymorphism (SNP) occurring in miRNA sequence or in the binding site of their target gene. It is involved in aberrant expression and maturation of these regulatory molecules. Up- or downregulated expression of miRNAs also results from regulation by defects of miRNA pathway or transcriptional factors for which their level is altered in affected cells [72,79]. It is also established that decreased miRNA expression can be the result of the action of other RNA molecules (long non-coding RNAs—lncRNAs, circular RNAs—circRNAs) that associate with them as a sponge [80,81,82,83,84].

### 3.2. Aberrant Expression of miRNA in Cervical Neoplasia and Cancer

In cervical lesions, increased oncomir expression and decreased tumor suppressors has been recorded in both cell/tissue cultures and clinical samples [85]. Furthermore, various miRNA molecules are correlated with advancement of intraepithelial neoplasia, HPV infection-related processes, or histological subtype [86,87]. Moreover, miRNAs are increasingly recognized as factors with predictive and prognostic potential for determining survival and metastatic risk in cancer patients. Various reports indicate that the level of certain miRNAs may be an indicator of a patient’s response to conventional treatment.

#### 3.2.1. Dynamics of miRNA Expression Observed in Cervical Lesions and Cancer

Over 250 significantly dysregulated miRNAs were identified in CC cell lines on the basis of microarray and real-time polymerase chain reaction (real-time PCR) assays. Substantially, miRNA tendency to be up- or downregulated was found to be similar in several studies, creating a possibility to establish a suitable diagnostic pattern [86,88,89,90,91]. miRNAs show altered expression that is dependent on the advancement stage of cervical lesion. Some species have an early transient character. This means that differential expression level is observed in CIN 2 and CIN 3 versus normal tissue. No significant differences occur with CC compared to normal tissue. Secondly, miRNA molecules may be altered continuously with increasing or decreasing expression level, beginning with early lesions. The highest changes are detectable between normal tissue and cancerous tissue. Late expression changes of miRNAs are identified in CC, and variability in normal tissue in comparison to cervical neoplasias is not significant. These dependencies were described by Wilting et al., presenting a detailed miRNA profile obtained by a microarray assay [86].

One pioneer study showed that overexpression of miR-15a, miR-26b, miR-195, miR-200c, and miR-223 and underexpression of miR-143 and miR-145 is significant in precancerous raft tissues (HPV infected). Then, miRNA profile was investigated in CC tissues, demonstrating compliancy with results obtained in rafts in the cases of miR-15a, miR-143, miR-145, and miR-223 expression levels. Moreover, miR-15b, miR-16, miR-126, miR-146a, miR-155, and miR-424 were significantly altered in CC tissues [85]. In other research, decreased expression of miR-10a, miR-26a, miR-29a, miR-99a, miR-143, miR-145, miR-196a, miR-199a, miR-203, and miR-513 and increased expression of miR-16, miR-27a, miR-106a, miR-142-5p, miR-197, and miR-205 was observed in the course of development of (pre)cancerous lesions [88]. McBee et al. demonstrated a profile of 10 differentiated miRNAs, including miR-16, miR-21, miR-106b, miR-135, miR-141, miR-223, miR-301b, miR-449a, miR-218, and miR-433 [92]. In another report, similar results to miRNA expression described earlier was confirmed, and altered expression was additionally demonstrated for upregulated miR-17, miR-20b, miR-25, miR-92a, and miR-224 and downregulated miR-10b, miR-34a, miR-100, miR-195, and miR-375 [89]. Cheung et al. found that 12 altered miRNAs: miR-9, miR-10a, miR-20b, miR-34b, miR-34c, miR-193b, miR-203, miR-338, miR-345, miR-424, miR-512-5p, and miR-518a may distinguish high-grade CIN from normal cervical tissues [93]. Rao et al. reported 18 upregulated and 19 downregulated miRNAs in CC biopsies. Among them, the highest fold changes (> 2, 5) compared to normal adjacent tissues was observed for oncomiRs miR-7, miR-20b, miR-31, miR-141, miR-142-5p, miR-200a, miR-224, and miR-429 and tumor-suppressive miR-10b, miR-99a, miR-100, miR-143, miR-145, miR-195, miR-218, miR-376, and miR-497 [94]. The results of further investigations provide strong evidence of the previously undertaken research validity, showing the same direction of changes in the miRNA profile depending on the stage of cervical pathology [87,90,91,95,96,97,98,99,100,101]. Results reported in cervical tissues that were repeated at least twice are summarized in Table 1.

#### 3.2.2. Differential Expression of miRNA between HPV-Positive and -Negative Cases

HPV infection plays a fundamental role for miRNA differential expression. Firstly, miRNAs are regulated by HR-HPV oncogenes, in particular E6/E7, through acting on p53 and pRB level. Due to this, all miRNAs regulated by p53 and pRB pathway signaling factors are considered to be modulated by these oncogenes. miR-218 expression is reduced in patients infected by HR-HPV and it decreases continuously on the basis of lesion advancement [109]. E7-encoded protein interaction with E2F leads to increased miR-15a and miR-16-1 expression level [110]. Panel of HR-HPV-dependent miR-34a, miR-21, miR-27a, miR-155, miR-203, and miR-221 has a significant prognostic value for HPV-positive SCC samples. However, the same set of miRNAs turned out to be non-indicative for HPV-positive AdCs [100]. Wang et al. reviewed miRNAs responding to E6/E7 in organotypic raft cultures by microarray and microRNA sequencing (miRNA-seq). Consistent results of responsive miRNAs was received for oncogenic miR-16, miR-25, miR-92a, miR-93, miR-106b, miR-210, miR-224, and miR-378 and tumor-suppressive miR-22, miR-24, miR-27a, miR-29, and miR-100. Most of them presented similar fold changes in CINs an CC tissues compared to non-infected culture [87]. Honegger et al. revealed significant upregulation of miR-143-3p, miR-23a-3p, miR-23b-3p, and miR-27b-3p and downregulation of miR-17-5p, miR-186-5p, miR-378a-3p, miR-378, miR-629-5p, and miR-7-5p after E6/E7 silencing in HeLa and SiHa cells, which was correlated with promoting cell growth, inhibiting apoptosis and senescence [76]. According to The Cancer Genome Atlas, the most aberrant expression between HPV-positive and -negative tissues is detectable for miR-944, miR-767-5p, and miR-105-5p [35]. Aberrant expression of miR-148a-3p, miR-199b-5p, and miR-190a-5p may be indicative for HPV-16 infection in cervical tissues [111]. Aberrant expression of four miRNAs investigated by Kawai et al. (miR-126-3p, miR-20b-5p, miR-451a, miR-144-3p) showed significant correlation with HPV-16/18 infection in clinical samples [104]. HPV-16 and HPV-18 oncogenes downregulate miR-375 expression in cervical cancer, promoting cell proliferation, migration, and invasion in CC cell lines [112]. Mandal et al. demonstrated that miR-16 is overexpressed in HPV-16-induced cervical cancer and downregulated in HPV-16-positive non-malignant tissues versus normal samples. miR-181c increased expression level is correlated with HPV-16-positive cervical cancers with episomal state of virus. miR-323 and miR-203 dysregulation is strongly associated with E7 expression, whereas downregulation of miR-34a, miR-143, miR-196b, and miR-203 is progressive to lesion advancement [63].

HPV frequently integrates into the part of the host genome where miRNA loci are localized. Altered expression of miR-9, miR-15b, miR-28-5p, miR-100, and miR-125b in cervical cancer tissues is established to be directly associated with chromosomal aberrations [86]. Other mechanisms are associated with E6/E7 overexpression that may lead to epigenetic modifications of miRNA’s promoter region or affecting important molecular pathways regulators.

Increasing importance in carcinogenesis is attributed to the occurring of E6 in several splice isoforms (E6*I, E6*II, E6*III, and unspliced E6fl), which has been described only for HR-HPV types [113]. Li et al. reported that miR-875 and miR-3144 impact on splice isoforms of E6 and can modulate its ratio directly or by the EGFR pathway [114].

Interestingly, several HPV-encoded miRNAs have been identified and validated [115,116], although they seem to have no diagnostic potential due to low expression in various cervical samples. However, they probably play important functions in the pathogenesis of infection, in particular in latency establishment and immune evasion [117].

In summary, the expression of several cellular miRNAs can be a significant indicator of HPV stability and determination of virus-induced aberrations of key cell cycle pathways.

#### 3.2.3. Variability in CC Histological Subtypes

Wilting et al. showed that 17 miRNAs exhibit significantly differential expression between SCC and AdC. Among them, eight were upregulated and nine downregulated. The highest significant fold changes (> 1,5) were observed in oncomiR’s expression, including miR-205, miR-222, miR-210, miR-27a, and miR-224. In the case of tumor-suppressive miRNAs, fold changes were lower. Reduced expression was the most significant for let-7g, miR-199b-5p, miR-215, miR-145, miR-194, and miR-375 [86]. Gocze et al. compared expression profiles between SCC and AdC with the same HR-HPV status. From seven miRNA analyses, they revealed an opportunity to distinguish two of the most common histological subtypes by miR-21, miR-27a, miR-34a, miR196a, and miR-221 [100]. The Cancer Genome Atlas Research Group study displayed upregulation of miR-375 and downregulation of miR-205-5p and miR-944. Furthermore, miR-99a-5p and miR-203a variability was determined for keratin-high and -low clusters of SCC [35]. Babion et al. tested eight candidate miRNA species to discriminate CC lesions. As a result, they obtained an unambiguous differentiation between SCC and AdC cases by analyzing different levels of miR-15b-5p/miR-375 expression [38].

#### 3.2.4. Prognostic Value—The Risk of Metastization

miRNA profiling, with a suitably selected set, may have a strong prognostic value in women affected by CC. It may verify advancement of disease and predict its development. According to several studies, some miRNAs’ altered expression level may increase the feasibility of poor clinical outcome. Fundamentally, the risk is associated with metastization and number of metastases resulting from the cell’s ability for proliferation, migration, invasion, and modulation of epithelial-to-mesenchymal transition (EMT) (Table 2). Dysregulation of metastamiRs correlates with FIGO (International Federation of Gynecology and Obstetrics) stage advancement and tumor size [118]. Upregulated in invasive CC miR-21 (detectable in body fluids), miR-20b, miR133b, miR-155, miR-205, and miR-499a are involved in a higher risk for lymph node metastasis and poor clinical stage. Overexpression of these miRNAs leads to higher proliferation, migration, colony formation, and invasion abilities by targeting genes associated with metastasis [119,120,121,122,123]. Furthermore, miR-205 and miR-20b influence the EMT processes and may become valuable therapeutic targets in the future [121,123].

miR-195, miR-375, miR-34a, miR-23b, miR-125a, and miR-223 downregulation is observed in invasive cervical cancer and leads to increased metastasis ability. Most are dependent from direct or indirect E6 action on the p53 regulation pathway. miR-375, miR-34a, and miR-23b may be silenced by promoter hypermethylation [112,126,127,128,129,131,132,137,138]. According to a recent study, overexpressed miR-21 and miR-16 contribute to downregulation of Kruppel-like factor 4 (KLF4) and estrogen receptor 1 (ESR1), which may be critical for invasive CC development [139]. Recently, Chen et al. identified a prognostic signature of four miRNAs—miR-502, miR-145 (downregulated), miR-142, and miR-33b (upregulated) in a data mining study. An established set is involved with high risk of lymph node metastasis in CC [140], although it is a model that requires further studies using clinical specimens.

#### 3.2.5. Prognostic Value—Susceptibility to Conventional Therapy

There is no doubt that a fundamental prognostic factor in the therapeutic process is to determine the applicability of treatment with chemotherapeutics or with the radiotherapy method. Recent studies performed in CC cell lines indicate that some miRNA dysregulated expression may be a factor specifying predictable response to cancer treatment (Table 3). It has been established that miR-155, miR-181a, miR-214, miR-497, miR-499a, and miR-664 are involved in the regulation of sensitivity to cisplatin [119,124,141,142,143,144]. miR-125a promotes sensitivity to paclitaxel, while upregulated miR-375 is associated with increased resistance to this chemotherapeutic [145,146]. Downregulation of miR-122-5p, miR-375, and miR-449b and upregulation of miR-21 and miR-125 are related to radioresistance of CC cells [147,148,149,150,151].

Determination of biomarkers contributing to metastasis ability and susceptibility to standard treatment may significantly increase the positive outcomes of the disease by individualized choice of effective therapeutic method. Moreover, the data suggest that the above miRNAs may be targets for the development of new therapeutic strategies for a more effective treatment of invasive CC.

Accordingly, the reports outlined above (Section 3.2.2, Section 3.2.3, Section 3.2.4, Section 3.2.5) attempt to select the best-studied miRNAs with the broadest known prognostic potential (Figure 1). The individual collections include miRNA species, the expression of which is (I) dependent on HPV oncogenes, (II) shows variability between the two most common histological types of CC, (III) determines the risk of metastasis, (IV) or may be related to compliance with conventional therapy. miRNAs included in each group are shown in Appendix A. As a result, four miRNA species with the widest known diagnostic and prognostic potential were identified. The most universal markers, included in all groups, seem to be continuously upregulated miR-21, and miR-375 downregulated in high grade lesions, followed by miR-155 (upregulated) and miR-34a (downregulated) that are common to three groups (I, III, IV, and I, II, III, respectively).

### 3.3. miRNAs as Biomarkers

It is assumed that the ideal cancer biomarker should create possibility and full reliability to distinguish patients affected by cancer from all patients without cancerous lesions. In the case of cervical cancer, it is also important to identify people from risk groups who have developed (or are presumed to develop) precancerous lesions (CIN 2–3). Therefore, the cervical pathology biomarker should have good discrimination rates for neoplastic lesions, characterized by high sensitivity and specificity of the determinations (both near 100%). When performing determination in blood samples, the biomarker should have tissue specificity in order to properly locate lesions. Great importance is also drawn to the possibility of predicting the probability of specific clinical conditions related to malignancy, such as the risk of lymph node involvement, local recurrence, distant metastases, or the patient’s possible reaction to treatment method that can be established on the basis of the level of biomarker. The cancer biomarker level should also have a high prognostic value, allowing for the assessment of the patient’s further fate after the implementation of treatment, for example, the assessment of the likelihood of asymptomatic or overall survival in cancer patients [152].

#### Standard Testing versus miRNAs as Cervical Pathology Biomarkers—Diagnostic Accuracy

In developed countries, screening tests are based on the detection of infection caused by HR-HPV, which is the main risk factor for the development of cervical (pre)cancerous lesions. HPV assays replace cytology-based primary screening or are utilized as a follow-up test for the patients with abnormal cytology results. However, the greatest level of efficacy is observed when both tests are performed simultaneously (co-testing), with this being the most recommended for primary screening [15].

Gynecological cytology is limited by low sensitivity of determination, ranging from 40% to 75% for precancerous lesion detection (specificity 84–96%) [153,154,155]. Currently, cytological preparations can be evaluated in two ways—as conventional Pap smear and as liquid-based cytology (LBC). Some comparative studies did not show significantly increased sensitivity or specificity of LBC; therefore, the clinical significance of the method selection is questionable [65,156,157]. Wentzensen et al. reported that sensitivity and specificity of LBC tests are 76.6% and 49.6% for CIN 2+ detection and 83.8 and 48.7% for CIN 3+ detection, respectively. Patients with established intraepithelial lesions, depending on the stage of their advancement, await re-examination (6–12 months) or undergo colposcopy and histopathology evaluation. Importantly, cytology testing alone does not detect a certain number of adenocarcinoma cases that are frequently observed in women who test negative for cytology and positive for HR-HPV infection [158,159]. HR-HPV test accuracy depends on the evaluation method, but essentially is characterized by higher sensitivity (69.1% vs. 40,6%) and lower specificity (94% vs. 97.3%) for CIN 3 detection in comparison to cytology [153].

Taking into account the imperfections of primary screening tests related to the risk of inconclusive results (cytology) or the inability to quickly assess the persistence of HPV infection, molecular tests discriminating high-grade squamous intraepithelial lesion (HSIL) from non-dysplastic lesions (negative for intraepithelial lesion or malignancy—NILM) and reducing the risk for misclassifications are approved for diagnostic use. p16INK4a and Ki-67 dual immunostaining in combination with cytology increase its sensitivity and specificity (83.4% vs. 76.6% and 58.9% vs. 49.6% respectively) [160]. Recently, an innovative test for HR-HPV-positive patients was introduced. The assay is based on multiplex quantitative polymerase chain reaction (qPCR) and indicates methylation levels of FAM19A4 and mir124-2 loci to predict the risk of CIN 3+ development. Simultaneous application of the test with HPV-16/18 genotyping achieves 88.5% sensitivity of CIN 3+ and is characterized by high prognostic value [161,162]. Despite the achievement of an increase diagnostic accuracy in the detection of lesions at risk of developing towards cancer, none of above assays meet the criteria for an ideal cervical pathology biomarker.

The promising compliance of miRNA expression in cell lines and biopsy material prompted researchers to look for new, less invasive solutions for the construction of diagnostic tests in the future. Tian et al., according to results obtained by Li et al. [106], constructed a panel of six miRNAs (miR-424, miR-375, miR-34a, miR-218, miR-92a, and miR-93) to analyze HPV-positive exfoliated cytological samples. Research revealed that the test exhibits higher potential for CIN 2+ and CIN 3+ discrimination than conventional cytology [103]. Ribeiro et al. analyzed miR-34a and miR-125b as potential invasive CC biomarkers, obtaining results proving good diagnostic utility for miR-125b. They suggest that this species may be a promising predictive and prognostic CC biomarker [108]. Kawai et al. analyzed 2588 miRNAs by microarray in biopsy cervical tissues and selected 22 upregulated candidates for validation by qPCR. From these, four oncomiRs with the highest fold changes observed in CIN 3 and CC versus normal samples—miR-126-3p, miR-20b-5p, miR-451a, and miR-144-3p—which were then investigated in cervical exfoliated cells. Utilization of these four miRNAs revealed high sensitivity and specificity in distinguishing NILM from precancerous and cancerous lesions, with the best performances for miR-451a and miR-144-3p [104]. The most promising study utilizing exfoliative cells was performed by Liu et al. They selected six miRNAs for testing as potential biomarkers in cervical lesions: miR-20a, miR-92a, miR-141, miR-183, miR-210, and miR-944. The best diagnostic performance for a single miRNA was observed for miR-183 in CIN discrimination, and for miR-141, which was the most effective to differentiate CC patients. For both CINs and CCs, the use of a combination of all six miRNA species improved diagnostic accuracy [163]. Table 4 presents the diagnostic parameters (sensitivity, specificity) obtained in studies on the use of miRNA as biomarkers in the development of cervical lesions. Reports using exfoliating cells have been taken into account.

Circulating miRNA profiling in serum samples also provides promising results and creates new opportunities to reduce invasiveness of sample collection. Jia et al. determined a valuable diagnostic set of five miRNAs overexpressed in CC in serum (miR-21, miR-29a, miR-25, miR-200a, miR-486-5p) identified by quantitative reverse transcription PCR (RT-qPCR) analysis. This test may be utilized in CC diagnosis with indication of histological grade and advancement stage [164]. Nagamitsu et al. found a significantly increased level of exosomal miR-483-5p, miR-1246, miR-1275, and miR-1290 in CC. Furthermore, they revealed that miR-29 may be a crucial element in miRNA network [165]. Xin et al. showed the potential for early detection of CINs by serum profile of four miRNAs: miR-9, miR-10a, miR-20a, and miR-196a. This panel displayed high accuracy in distinguishing CINs and correlation with HPV infection status [166]. A recent study established that miR-30d-5p and let-7d-3p may have diagnostic utility to screen CC and cervical neoplasia. However, equivocal tissue and serum miRNA profiles were obtained. Presumably, it is an effect of selective secretion of miRNAs in exosomes and specific tumor environment interaction [167]. Exosomal miRNA species may not be tissue-specific, meaning that different processes of any location may cause fluctuations in serum of miRNA expression level.

## 4. Summary and Conclusions

Cervical cancer continues to be at the forefront of mortality among middle-aged women, as evidenced by cancer statistics [5,8]. As most cervical cancers, especially in developing countries, are detected at an advanced stage, exploiting the potential of miRNAs as prognostic factors and as indicators of treatment susceptibility is well-founded. An effective screening of women from risk groups also remains extremely important.

miRNA, despite the intensification of research on their functionality and clinical utility, are still relatively poorly understood. However, it is known that their dynamic expression changes are associated with key determinants of neoplastic diseases [72]. Their impact on series of cell signaling pathways related to tumorigenesis creates a wide research field towards their use as diagnostic and prognostic (pre)cancer biomarkers [73,100,118,140,163]. miRNAs are relatively easy to measure under laboratory conditions due to the much greater stability of the molecules compared to other RNA fractions [168]. However, the techniques of isolation, determination, normalization, and analysis of results require standardization in order to obtain repeatability and comparability of test results obtained in various research centers. This comparability is also not fully effective due to different study designs; different criteria for selecting the reference group; not taking into account other cancer risk factors; and the individual variability, which remains extremely difficult to reliably assess.

Expression levels of miRNAs at different stages of cervical pathology usually fluctuate continuously, or statistically significant changes appear only at an advanced stage of cancer progression [86]. Moreover, the variation ranges of the expression level may overlap to a large extent, which can be problematic in establishing clear cut-off values. As presented in Table 1, the best confirmed differential miRNA species in cervical lesions noted in the most of reports are upregulated miR-21, miR92a (early continuous increasing), miR-9, miR-15b, miR-16, miR-20b, miR-141, and miR-155 (late expression) and downregulated miR-100, miR-195, miR-203, miR-375, miR-424 (continuous decreasing of expression), miR-34a, miR-99a, and miR-125 (reduced commonly in cervical cancer). Moreover, the demonstrated discrepancies in the dynamics of changes in miRNA expression may indicate earlier disturbances in the cell balance, at a time when cytological/histological changes are still elusive. Such a feature, if confirmed in future research, could represent a significant advantage in terms of high predictive value of potential biomarkers.

Variation of the expression of miRNA is multifactorial [22]. Therefore, the search for one miRNA as the perfect marker seems to be a rather limited approach. It seems more reasonable to create a miRNA signature that takes into account most of these factors, corresponding to the patient’s clinical condition, which could be a specific fingerprint of cervical pathology [100,164]. As described in Liu et al., a pattern of six miRNA species (miR-20a, miR-92a, miR-141, miR-183, miR-210, miR-944) achieved great diagnostic performance in differentiating cervical (pre)cancerous lesions [163]. Therefore, combination of several miRNAs can significantly improve the diagnostic accuracy of the assays, and, with the establishment of clear criteria for the results’ interpretation, provide an efficient way to discriminate between various precancerous conditions. Furthermore, miRNA profile can be determined from the same sample as cytology and HR-HPV test, namely, from exfoliated cells noninvasively collected on the LBC medium [103,104,108,163].

Consideration of predictive and prognostic factors seems to be a big challenge, as there are still not enough reports of extensive studies in the reference group of women with various cervical pathological conditions. However, researchers are getting closer to establishing the relationship between miRNA expression levels and sensitivity to conventional methods of cervical cancer treatment that requires confirmation of results received in CC cell lines.

In view of heterogeneity of CC molecular features, it should not be treated as a single disease, both in the diagnostic and therapeutic process, and requires a personalized approach. Introducing new diagnostic solutions for personalized medicine is challengeable and requires accurate knowledge about a cell’s dysregulation at the molecular level.

miRNA profiling may provide a detailed fingerprint of a cell’s condition and, in reference to current reports, seems to have high potential to be the marker-determining multifactorial process with relation to cervical neoplasia and cancer development, having strong predictive and prognostic value. However, the satisfactory preliminary results of scientists in this matter requires confirmation in standardized, extensive clinical trials conducted in experienced research centers to develop clear determination algorithms and introduce a new, broad-spectrum and noninvasive biomarker into the clinical use.

## Figures and Tables

**Figure 1 jcm-09-03668-f001:**
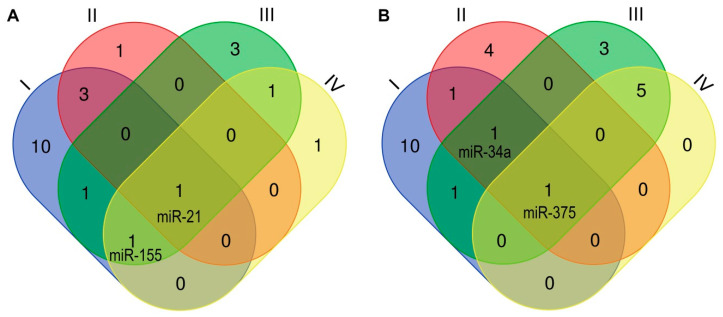
Venn diagram showing upregulated (**A**) and downregulated (**B**) miRNAs depending on human papillomavirus (HPV) infection (I), histological type of cervical cancer (CC) (II), the risk of metastasis (III), and susceptibility to therapy (IV).

**Table 1 jcm-09-03668-t001:** Dynamics of microRNA (miRNA) expression revealed in cervical biopsies or exfoliative cells.

	Early Transient [References]	Early Continuous [References]	Late [References]
Overexpressed	miR-10a [86,93]	miR-10a [88]	**miR-9** [86,90,93,95,102]
	miR-34b [93]	miR-34b [86]	miR-15a [86,90]
		miR-28 [86,102]	**miR-15b** [85,86,89,90,102]
		**miR-92a** [86,87,89,103]	miR-17 [86,89,90]
		miR-93 [89,103]	**miR-20b** [86,89,90,93,94,104]
		**miR-16** [92]	**miR-16** [85,86,87,88,89,90,97]
		**miR-21** [86,92,98,105]	**miR-21** [90,95,97]
		miR-25 [86]	miR-25 [87,89]
		**miR-** **141** **[92]**	**miR-** **141** **[86,90,94]**
		**miR-155** [89]	**miR-155** [85,86,95]
		miR-196a [88,96]	miR-27a ^1^ [87,88]
		miR-200a [86,94,97]	miR-31 [90,94,99]
			miR-106a [86,88,89]
			miR-106b [90,92,97]
			miR-124 [91,92]
			miR-142-5p [88,94]
			miR-223 [85,92,97]
			miR-224 [89,94]
Underexpressed	**miR-100** [87]	**miR-100** [89,94,106]	**miR-100** [86,102]
		miR-29a ^1^ [87,88,89]	miR-10b [89,94]
		**miR-34a ^1^** [89,103]	**miR-34a ^1^** [100,107,108]
		**miR-99a** [88,89]	**miR-99a** [86,94,97]
		**miR-125b** [108]	**miR-125b** [86,89,97,102]
		**miR-195** [89,90,94]	**miR-195** [86]
		miR-199a [88]	miR-199a [86]
		**miR-** **218** **[89,90,92,103,109]**	**miR-218** [86,91,94,107]
		**miR-375** [89,101,103]	**miR-375** [86,102]
		miR-497 [90,94]	miR-497 [86]
		miR-513 [88]	miR-513 [86]
		miR-143 ^1^ [88,94]	miR-143 ^1^ [85]
		miR-145 [88]	miR-145 [85]
		**miR-** **424** **[89,93,103]**	**miR-424** [85]
		miR-145 ^1^ [88,89,94]	miR-376 [86,90,94]
		miR-149 [86,102]	
		miR-193b ^2^ [86,97]	
		**miR-203** [86,88,97,100,102]	

^1^ miRNA showing the opposite direction in Wilting et al. [86], ^2^ miRNA showing the opposite direction in Jimenez-Wences et al. [91]. Marked in color—miRNAs showing the same direction in close related stages of cervical pathology; bold—miRNAs showing the same direction in at least four cited reports.

**Table 2 jcm-09-03668-t002:** Impact of metastamiRs contributing to cervical cancer.

	miRNA	Target	Processes Related to Metastization	Reference
Overexpressed	miR-20b	TIMP metallopeptidase inhibitor 2 (TIMP2)	Migration, invasion, EMT modulation	[123]
	miR-21	RAS p21 protein activator 1 (RASA1)	Migration, invasion	[122]
	miR-133b	Mammalian sterile 20-like kinase 2 (MST2), Cell division control protein 42 homolog (CDC42), Ras homolog gene family member A (RHOA)	Proliferation, colony formation	[120]
	miR-155	Tumor protein p53 (TP53), mothers against decapentaplegic homolog 2 (SMAD2), cyclin D1 (CCND1) Epidermal growth factor (EGF)	Migration, invasion EMT modulation	[119]
	miR-205	Connective tissue growth factor (CTGF), cysteine-rich angiogenic inducer 61 (CYR61)	Proliferation, migration	[121]
	miR-499a	Sex-determining region Y-box 6 (SOX6)	Proliferation, migration, invasion	[124]
	miR-944	HECT domain ligase W2 (HECW2), S100P-binding protein (S100PBP)	Proliferation, migration, invasion	[125]
Underexpressed	miR-23b	Urokinase plasminogen activator (uPA)	Migration, EMT modulation	[126]
	miR-34a	Notch receptor 1 (NOTCH1), jagged canonical notch ligand 1 (JAGGED1), E2F transcription factor 3 (E2F3)	Invasion, EMT modulation	[127] [128]
	miR-125a	Signal transducer and activator of transcription 3 (STAT3) Microtubule affinity regulating kinase 1 (MARK1), ABL proto-oncogene 2, non-receptor tyrosine kinase (ABL2)	Migration, proliferation, EMT modulation	[129] [130] [131]
	miR-195	Cyclin D1, mothers against decapentaplegic homolog 3 (SMAD3)	Proliferation, migration, invasion	[132] [133]
	miR-218	Laminin subunit beta 3 (LAMB3), baculoviral IAP repeat-containing 5 (BIRC5), Scm-like with four Mbt domains 1 (SFMBT1), defective in cullin neddylation 1 domain-containing 1 (DCUN1D1)	Migration, invasion, EMT modulation	[134] [135] [136]
	miR-223	Forkhead box O1 (FOXO1)	Proliferation, EMT modulation	[137]
	miR-375	Sp1 transcription factor (SP1), Astrocyte elevated gene-1 (AEG-1)	Proliferation, migration, invasion, EMT modulation	[138] [112]

**Table 3 jcm-09-03668-t003:** Impact of dysregulated miRNA on standard cervical cancer (CC) treatment.

	miRNA	Target	Response to Treatment	Reference
Overexpressed	miR-21	Phosphatase and tensin homolog deleted on chromosome 10 (PTEN)	Radioresistance	[148]
	miR-125	p21	Radioresistance	[149]
	miR-181a	Protein kinase C delta (PRKCD)	Resistance to cisplatin	[142]
	miR-375	Not specified	Acquired resistance to paclitaxel	[145]
	miR-499a	SRY-box transcription factor 6 (SOX6)	Resistance to cisplatin	[124]
Underexpressed	miR-122-5p	Cell division cycle 25A (CDC25A)	Radioresistance	[151]
	miR-125a	Signal transducer and activator of transcription 3 (STAT3)	Resistance to cisplatin and paclitaxel	[146]
	miR-155	Epidermal growth factor (EGF) SMAD family member 2 (SMAD2) Cyclin D1 (CCND1)	Resistance to cisplatin	[119]
	miR-214	BCL2-like 2 (Bcl2L2)	Resistance to cisplatin	[141]
	miR-375	Ubiquitin protein ligase E3A (UBE3A)	Radioresistance	[147]
	miR-449b-5p	Forkhead box P1 (FOXP1)	Radioresistance	[150]
	miR-497	Transketolase (TKT)	Resistance to cisplatin	[144]
	miR-664	E-cadherin	Resistance to cisplatin	[143]

**Table 4 jcm-09-03668-t004:** Diagnostic accuracy of miRNA-based exfoliative cytology testing in differentiating of cervical lesions.

Clinical Stage	Tested miRNA	AUC	Sensitivity (%)	Specificity (%)	Reference
LSIL	miR-451a miR-144-3p	0.850 0.850	76.0 68.0	82.0 89.0	[104]
	**miR-183** **6 miR pattern**	**0.990** **0.998**	**95.0** **97.9**	**97.0** **98.6**	[163]
HSIL	miR-451a miR-144-3p	0.870 0.870	80.0 75.0	82.0 88.0	[104]
	miR-424 miR-424/375/218	0.840 0.874	76.0 74.4	78.1 85.3	[103]
	**miR-183** **6 miR pattern**	**0.980** **0.996**	**92.0** **97.2**	**92.0** **96.6**	[163]
CC	miR-451a miR-144-3p	0.940 0.930	83.0 87.0	91.0 89.0	[104]
	miR-125b	0.802	88.0	69.0	[108]
	**miR-141** **6 miR pattern**	**0.942** **0.959**	**82.8** **91.4**	**91.7** **87.6**	[163]

Bold—trials with the best diagnostic performance. Abbreviations: AUC—area under the curve (receiver operating characteristic (ROC) analysis, LSIL—low-grade squamous intraepithelial lesion, HSIL—high-grade squamous intraepithelial lesion, CC—cervical cancer.

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
