# Peer review of "MicroRNA-Based Fingerprinting of Cervical Lesions and Cancer"

_jcm, 2020, doi:10.3390/jcm9113668_

Round 1
Reviewer 1 Report
Overall, this is a well-written manuscript that includes both conceptual information on HPV-induced cancers as well as the role of miRNA in this process, with a focus on possible biomarkers. The literature review is relatively comprehensive, and the compilation of results reported here is likely to be useful to researchers in this field.
Some specific comments are noted below:
- The review reads more like a collection of data than an integration and summation of the most important points. For example, it would be helpful to have a section near the end listing the most promising species or combination of species that might be used for developing diagnostic and/or therapeutic approaches.
- For Table 1, the relevant references should be somehow indicated. Of particular interest would be those species that are identified in the majority of the publications referenced. For example, is there a way to highlight or color those that showed up in, say, more than 50% of the references?
- Paragraph 2.2.2 starting from line 162, and the reference 53 (Hu et al, 2015) – it might be worth noting that the year following this publication, a letter to the journal claimed that the majority of the sites were artifact (https://www.nature.com/articles/ng.3392).
Author Response
Dear Reviewer,
thank you for your valuable comments. We appreciate you having carved out of your schedules to evaluate our manuscript for free.
We have improved the Manuscript as much as it was possible. We also respond to further comments below.
Point 1: The review reads more like a collection of data than an integration and summation of the most important points. For example, it would be helpful to have a section near the end listing the most promising species or combination of species that might be used for developing diagnostic and/or therapeutic approaches.
Response 1: We thank the reviewer for pointing this out. We have supplemented it, as follows:
- in the revised Table 1 (pages 8-10 of the revised manuscript) the best known miRNA species showing the defined dynamics of changes in expression are distinguished in bold
- in section 3.2.5. (Lines 449-458, page 15) we inserted brief analysis to present the best known miRNA species with the broadest diagnostic potential; a diagram illustrating this selection has been inserted below (page 16)
- The new subsection has been separated (lines 463-550, pages 16-18 of the revised manuscript). It presents miRNAs (single and combination of species) for which diagnostic accuracy was determined in non-invasive cytology specimens (Table 4, page).
- We attached the summary, which highlights the best known miRNAs and those with application potential in the non-invasive diagnostics of cervical neoplastic lesions (Lines 569-591).
Point 2: For Table 1, the relevant references should be somehow indicated. Of particular interest would be those species that are identified in the majority of the publications referenced. For example, is there a way to highlight or color those that showed up in, say, more than 50% of the references?
Response 2: This part has been fully rewritten and supplemented with additional data from the literature, missing before (lines 272-342; Table 1 – pages 8-10).
The data contained in the table has been revised in detail. After correction, it contains repeatable reports on the direction of changes in the expression of miRNA, in at least two positions.
It is quite difficult to compare individual publications due to the different study designs - some reports examine entire panels, and some focus only on one species of miRNA. Therefore, the ones that were mentioned at least 4 times are bolded.
The data in Table 1 was also organized in terms of the dynamics of expression changes (marked in color) and commented in the summary (Lines 572-580).
Point 3: Paragraph 2.2.2 starting from line 162, and the reference 53 (Hu et al, 2015) – it might be worth noting that the year following this publication, a letter to the journal claimed that the majority of the sites were artifact (https://www.nature.com/articles/ng.3392).
Response 3: We have removed above reference from the bibliography without the need to change the text. We apologize for the oversight and thank you for pointing this out.
We would like to thank the referee again for taking the time to review our manuscript.
Sincerely,
Authors
Reviewer 2 Report
The review manuscript entitled “MicroRNA - based fingerprinting of cervical 2 pathology “ by Pisarska and Baldy-Chudzik intended to systematize current reports on miRNA capacity that can be utilized in personalized diagnostics of cervical precancerous and cancerous lesions.
One of the major statements of this manuscript is “To date, no standardized diagnostic algorithm using the miRNA pattern in cervical pathology has been defined. However, the high sensitivity and specificity of the reported assays gives hope for the development of non-invasive diagnostic tests that take into account the heterogeneity of tumor-related changes.” However, the data as presented do not clearly support the statement. The authors need to present these specificities and sensitivities and to compare them with those of Pap-smear and HR-HPV tests, as well as other gene tests. What are the real advantages of miRNA markers is unclear. In addition, the review is written as a body of collection of existing data without further analyses of potential connections, such as a pathway analysis. Moreover, the title of the review paper may be changed, since more than 1/3 of the review is on CC in general, not on miRNAs. The Conclusion should be expanded, since there are many issues/limitations needed to be addressed before moving miRNA into clinic application for CC, which should be more carefully discussed.
Author Response
Dear Reviewer,
thank you for your valuable comments. We appreciate you having carved out of your schedules to evaluate our manuscript for free.
We have improved the Manuscript as much as it was possible. We also respond to further comments below.
Point 1: One of the major statements of this manuscript is “To date, no standardized diagnostic algorithm using the miRNA pattern in cervical pathology has been defined. However, the high sensitivity and specificity of the reported assays gives hope for the development of non-invasive diagnostic tests that take into account the heterogeneity of tumor-related changes.” However, the data as presented do not clearly support the statement. The authors need to present these specificities and sensitivities and to compare them with those of Pap-smear and HR-HPV tests, as well as other gene tests.
Response 1: We agree with the Reviewer that it was necessary to develop this issue. The new subsection entitled “3.3. miRNAs as biomarkers” has been separated (lines 463-550, pages 16-18 of the revised manuscript).
The subsection fully describes the sensitivity and specificity of the screening tests commonly used in diagnostics (pap-test, HR-HPV-test, p16INK4A / Ki67 dual immunostaining, FAM19A4 / mir-124-2 methylation test) (lines 478-507). Then, the review of the literature data on miRNA expression analysis with regard to pathological changes of the cervix found in non-invasively collected clinical material (Line 508-550) was transferred (from lines 315-336). Data requiring clear visualization, i.e. promising high diagnostic parameters of miRNAs, are included in the newly created Table 4. Tests performed on exfoliated cells were taken into account, because in our opinion it is a more reliable diagnostic material than in the case of blood serum and exosomal miRNAs, which, however, are subject to selective secretion and are not always tissue specific. In addition, the same material can be used to study the miRNA profile as for cytology (LBC) and HR-HPV tests.
Point 2: The review is written as a body of collection of existing data without further analyses of potential connections, such as a pathway analysis.
Response 2: We appreciate the reviewer’s suggestion and agree that it would be useful to demonstrate potential connections. However, a pathway analysis is beyond the scope of our paper, which aims only to systematize miRNA capacity that can be utilized in personalized diagnostics of cervical precancerous and cancerous lesions.
A researcher of the relationship of miRNAs with cervical lesions and cancer may have great difficulties in selecting a huge information base, and in addition, it is growing at an extremely fast pace.Therefore, we believe that the article should distinguish a groups of miRNAs that are best known and show high capable diagnostic potential (including predictive and prognostic).We have integrated it, as follows:
- in the revised Table 1 (pages 8-10 of the revised manuscript) the best known miRNA species showing the defined dynamics of changes in expression are distinguished in bold
- in section 3.2.5. (Lines 449-458, page 15) we inserted brief analysis to present the best known miRNA species with the broadest potential; a diagram illustrating this selection has been inserted below (page 16)
- We attached the summary, which highlights the best known miRNAs and those with application potential in the non-invasive diagnostics of cervical neoplastic lesions (Lines 569-591).
Point 3: The title of the review paper may be changed, since more than 1/3 of the review is on CC in general, not on miRNAs.
Response 3: The change of the title has been done, but only to clarify the pathological changes referred to in the review.
With this and the previous remark in mind, the miRNA section has been expanded and many previously unclear issues have been corrected.
We were not going to outgrow the features of cervical cancer. In our opinion, however, only the necessary information on this pathology were provided. Section 2 describing this condition is intended to bring the reader the essential information and could not be omitted, due to the subsequent presentation of the relationship between the pathogenesis of cervical cancer and miRNA aberrations occurring in various stages of cancer-related lesions. This section covers only 2.5 pages, including the characteristics of the main causative agent (HPV), which could not be missing from review on such topics.
We hope that the title reflects the nature of this review more closely, following these changes that we have been applied.
Point 4: What are the real advantages of miRNA markers is unclear. The Conclusion should be expanded, since there are many issues/limitations needed to be addressed before moving miRNA into clinic application for CC, which should be more carefully discussed.
Response 4: We thank the Reviewer for pointing this out. We supplemented the article with a summary preceding conclusions in which we discuss both the benefits of potential miRNA-based testing and its limitations in introducing into clinical use (lines 551-596; page 19). The conclusions were also clarified (lines 601-612; page 20).
We would like to thank the referee again for taking the time to review our manuscript.
Sincerely,
Authors